# Towards Bioprospection of Commercial Materials of *Mentha spicata* L. Using a Combined Strategy of Metabolomics and Biological Activity Analyses

**DOI:** 10.3390/molecules27113559

**Published:** 2022-05-31

**Authors:** Juan Camilo Henao-Rojas, Edison Osorio, Stephanie Isaza, Inés Amelia Madronero-Solarte, Karina Sierra, Isabel Cristina Zapata-Vahos, Jhon Fredy Betancur-Pérez, Jorge W. Arboleda-Valencia, Adriana M. Gallego

**Affiliations:** 1Corporación Colombiana de Investigación Agropecuaria-Agrosavia, Centro de Investigación La Selva, Kilómetro 7, Vía a Las Palmas, Vereda Llanogrande, Rionegro 054048, Colombia; imadronero@agrosavia.co; 2Grupo de Investigación en Sustancias Bioactivas GISB, Facultad de Ciencias Farmacéuticas y Alimentarias, Universidad de Antioquia, Cl. 70 No. 52-21, Medellin 0500100, Colombia; edison.osorio@udea.edu.co (E.O.); karina.sierra@udea.edu.co (K.S.); 3Hierbas y Plantas Tropicales SAS-HIPLANTRO, Cra. 56a No. 72a 101, Itagüí 055410, Colombia; estefania@hiplantro.com; 4Facultad de Ciencias de la Salud, Atención Primaria en Salud, Universidad Católica de Oriente, Rionegro 054040, Colombia; izapata@uco.edu.co; 5Centro de Investigaciones en Medio Ambiente y Desarrollo—CIMAD, Facultad de Ciencias Contables, Económicas y Administrativas, Universidad de Manizales, Cra. 9 No 19-03, Manizales 170001, Colombia; jbetancur@umanizales.edu.co (J.F.B.-P.); jwilliam.arboleda@udea.edu.co (J.W.A.-V.); 6Grupo de Investigación FITOBIOL, Instituto de Biología, Facultad de Ciencias Exactas y Naturales, Universidad de Antioquia, Cl. 67 No 53-108, Medellin 050010, Colombia; 7Biomasnest, Medellin 050010, Colombia

**Keywords:** spearmint, commercial materials, antimicrobial activity, antioxidant activity, untargeted metabolomics, bioprospection

## Abstract

Spearmint (*Mentha spicata* L.) has been widely studied for its diversity of compounds for product generation. However, studies describing the chemical and biological characteristics of commercial spearmint materials from different origins are scarce. For this reason, this research aimed to bioprospecting spearmint from three origins: Colombia (Col), Mexico (Mex), and Egypt (Eg). We performed a biological activity analysis, such as FRAP, DPPH, and ABTS, inhibition potential of *S. pyogenes*, *K. pneumoniae*, *E. coli*, *P. aeuroginosa*, *S. aureus*, *S aureus* Methicillin-Resistant, and *E. faecalis.* Furthermore, we performed chemical assays, such as total polyphenol and rosmarinic acid, and untargeted metabolomics via HPLC-MS/MS. Finally, we developed a causality analysis to integrate biological activities with chemical analyses. We found significant differences between the samples for the total polyphenol and rosmarinic acid contents, FRAP, and inhibition analyses for Methicillin-Resistant *S. aureus* and *E. faecalis*. Also, clear metabolic differentiation was observed among the three commercial materials evaluated. These results allow us to propose data-driven uses for the three spearmint materials available in current markets.

## 1. Introduction

Approximately 25–30 Mentha species have been reported worldwide, including *Mentha piperita* L., *Mentha arvensis* L., *Mentha suaveolens* Ehrh, and *Mentha spicata* L., stand out as some of the most representative of the *Lamiaceae* family. Among them, spearmint (*Mentha spicata* L.) has presented an increase in worldwide cultivation, a higher recognition for its intense aroma, and more reports regarding its stimulant, diaphoretic, antiseptic, gastrointestinal respiratory, and antispasmodic effects [1,2,3,4]. The producing regions are concentred in the United States, Egypt, Australia, and some areas of Asia. Recently, Mentha production and demand have increased in Latin American countries, such as Mexico and Colombia [5].

The commercialization of *Mentha spicata* L. as fresh leaves is common in restaurants and haute cuisine. As dried-ground leaves, it is used at the agro-industrial level in the production of extracts for the pharmaceutical, cosmetic, confectionery, functional foods, toothpaste, or herbal infusions industries. Additionally, spearmint extracts have recently been shown to possess antibacterial, antifungal, antiviral, insecticidal, and antioxidant properties [6,7,8].

Some authors have reported spearmint compounds with multiple applications in the generation of functional foods and personal care products [9], primarily phenolic acids such as protocatechuic acid, hydroxybenzoic acid, 4-hydroxy cinnamic acid, caffeic acid, syringic acid and ferulic acids, gallic acid, vanillic acids, p-coumaric acid, and rosmarinic acids [10]. For similar applications, flavonoids such as thymonin, naringenin, rutin, quercetin, esculetin, nodifloretin, luteolin, and scopoletin have also been reported [10].

A considerable amount of research has been carried out on the identification, biochemical characterization, localization, and health benefits of the metabolites of spearmint [11,12]. However, few studies have focused on evaluating the biological properties, quality, and chemical diversity of spearmint’s commercial materials present in the international market [3,7]. Materials from different countries were chosen because they retain the highest uses by food and cosmetic companies in the current Colombian market. Its proper use could generate high rates of productive performance, efficiency, and homogeneity of the products obtained, impacting direct economic costs.

The present work focused on identifying the main chemical properties and biological activities of commercial spearmint extracts grown in three different origins of Colombia (Col), Mexico (Mex), and Egypt (Eg). Our results provide tools for data-driven decision-making in selecting specific raw materials of *Mentha spicata* L. for the future generation of products.

## 2. Results

### 2.1. Chemical Measurements of Mentha spicata *L.*

The TPC ranged from 1183 to 1781 mg GA/100 g extract (Figure 1a), and significant differences were presented among the samples. Mexico had the highest TPC value, followed by Eg and then Col. The content of rosmarinic acid (RA) was higher in Eg, followed by Mex, and next by the Col samples. Eg showed a significant difference in RA compared to the other two origins (Figure 1b).

### 2.2. Biological Activity Assays

#### 2.2.1. Antioxidant Activities

The results of the antioxidant activity for the spearmint extracts are shown in Figure 2. The reductive capacity FRAP of Mex (Figure 2a) presented a significantly higher value of 1963.3 ± 76.5 mg eq of ascorbic acid (AA)/100 g than the other two origins. In addition, although no sample showed significant differences for the ABTS analysis, Mex showed the highest inhibition percentage (27.4 ± 2.0 mmol eq TX/100 g) (Figure 2b). Similarly, we did not find significant differences between the samples; however, Eg showed the highest value of 2,2-diphenyl-1-picrylhydrazyl (DPPH) among the samples (13.1 ± 2.3 mmol eq TX/100 g) (Figure 2c).

#### 2.2.2. Antimicrobial Activities

The antimicrobial activity assay showed that extracts obtained from different commercial materials of Mentha spicata inhibit the growth of Staphylococcus aureus Methicillin-resistant (SARM) (ATCC 43300) and Enterococcus faecalis (ATCC 19433) but did not show effects on the other tested bacteria. After 24 h of incubation, the growth of SARM showed a growth reduction of 53% and 24% with Col and Mex extracts, respectively. The Eg extract did not affect SARM’s growth (Figure 3a). In addition, the extract obtained from Colombian plants resulted in a more stable reduction of both bacteria compared with the positive control. Likely, E. faecalis was reduced by 56%, 68%, and 91% for Col, Eg, and Mex extracts, respectively (Figure 3b). Furthermore, we evaluated other gram-negative bacteria (Escherichia coli, Klebsiella pneumoniae, Pseudo-monas aeruginosa, and Streptococcus pyogenes), which did not show inhibition in spearmint extracts. Altogether, the results show the promising antimicrobial activity of M. spicata commercial extracts against gram-positive bacteria, such as SARM and E. faecalis, which are among the most significant human bacterial pathogens in clinical medicine.

### 2.3. Metabolomic Analysis

The metabolic profiling for the spearmint samples from the three different origins of Col, Eg, and Mex was analyzed using an untargeted approach via HPLC-MS/MS. The Partial Least Squares Discriminant Analysis (PLS-DA) was carried out to assess the discriminatory and predictive ability of the metabolite profiles among the three spearmint origins. Also, PLSD-DA was used to distinguish samples of different geographic origins by chemical composition. The PLS-DA plot showed that all biological replicates for each origin were grouped together, suggesting the consistency of our experimental groups (Figure 4a). Also, the Col samples were further separated from Eg and Mex origins and showed a low metabolic variability within samples compared to Eg and Mex. PC1 and PC2 accounted for 85.3% of the total variation. The PLS-DA was validated using the leave one cross-validation algorithm (LOOCV) to determine the model’s performance. A high predictive accuracy (R^2^Y = 0.837) was found; the exogenous variables (origin) explained the endogenous variation of the dependent variable. Likewise, the model’s predictive power (Q^2^ = 0.738) is considered reliable. Subsequently, we were able to tentatively automatically annotate 932 metabolites on our HPLC/MS data (Appendix A).

Furthermore, about 72 features were selected for further fragmentation and annotation via MS/MS using the Variable Importance Parameter (VIP) value > 2.0 criteria. Then, we manually annotated six metabolites using an in-house database built for mint, which is represented in the loading plot (Figure 4b). We tentatively annotated one metabolite of each chemical class of cycloparaffin, flavone, carotenoid, hydroxycoumarin, cinnamate ester, and caffeic acid ester (Table 1 and Appendix A). Esculetin contributed to the separation of Col, the rosmarinic acid separated Eg from the other two origins, and the rest of the metabolites were grouped for the Mex origin.

Next, we plotted a heatmap for the six annotated metabolites to display differences in their expression among the three spearmint origins (Figure 5). The clustering analysis allowed us to identify two groups agreeing with the PLS-DA. Most metabolites showed a contrasting pattern in Col (C1) compared to the Eg and Mex origins (C2). Most of the metabolites showed a low expression in Col compared to the other two origins. Furthermore, we performed a correlation heatmap for these six metabolites. Interestingly, esculetin was negative and significantly (*p*-value < 0.05) correlated with the rest of the metabolites (Appendix A and Appendix A).

### 2.4. Causality Analysis

A causality analysis has an approach that allows us to understand, in a multidimensional way, how a set of predictor variables influence a response variable. A causality analysis differs from a correlation analysis in the former, which considers the sum of the variables and is not based on pair-wise relationships. A causality analysis is helpful because, in a multi-component matrix, some can produce synergy between them while others act oppositely. By performing this analysis, we were able to discriminate which metabolites and in what proportion influence the different biological activities studied.

The results show that there are predictors (e.g., chemical analysis or specific metabolites) generating positive or negative causal relationships against the antioxidant activity. Here, a positive causal relationship means a predictor favors the antioxidant activity. In contrast, a negative causal relationship means a predictor disfavor of the antioxidant activity. TPC and the metabolites, chlorogenic acid and 7-methylene-1,3,5-cyclooctatriene, contributed to the antioxidant capacity, while esculetin and salvigenin did not contribute to the antioxidant capacity (Figure 6a). This is in concordance with reports where a positive correlation between antioxidant activities and TPC and chlorogenic acid has been reported [13,14,15].

Regarding the antimicrobial activities, the chlorogenic acid, esculetin, and 7-methylene-1,3,5-cyclooctatriene showed a positive contribution to the increase of the SARM inhibitory capacity. Contrary, the metabolites anteraxanthin, rosmarinic acid, salvigenin, and TPC, showed a negative influence on SARM’s inhibitory capacity. A different case was presented for the inhibitory activity of *E. faecalis*. Here, the chlorogenic acid, 7-Methylene-1,3,5-cyclooctatriene, and TPC increased the inhibitory capacity for this microbial strain. In contrast, esculetin, rosmarinic acid, and anteraxanthin reduced the effectiveness in the inhibitory activity of *E. faecalis* (Figure 6b).

The causality analysis results indicate that it is generally possible to obtain the predictive capacity of the antioxidant and antimicrobial activities of the samples of Mentha spicata extracts based on their discriminant metabolites. The above modified polynomial models with an adjusted R square greater than 98.5.

We further searched by uses reported in the literature for the annotated metabolites (Table 2). We found uses reported for antheraxanthin, salvigenin, rosmarinic acid, esculetin, and chlorogenic acid, except for 7-Methylene-1,3,5-cyclooctatriene. Whereas antheraxanthin reports cosmetical uses, the rest of the metabolites are reported to have the potential for health-related uses.

## 3. Discussion

This study provides a characterization of the antioxidant and antimicrobial activities and chemical composition (using chromatographical and spectrometry) for commercial spearmint of the three origins of Col, Mex, and Eg. Studying the bioprospection of the currently available commercial spearmint materials is relevant because it allows the mint market companies to adjust their portfolios and make better decisions using data-driven strategies.

With respect to antioxidant activities, the DPPH free radical inhibition capacity is consistent with reports from the Oman region (Arabian Peninsula), which reported 54.68% inhibition. In this study, the authors associated the inhibition mainly with the presence of terpenes in the extracts of spearmint [27]. Furthermore, our three commercial extracts exhibited a higher reducing capacity measured by FRAP than Yousuf et al. [28]. The high capacity to trap free radicals is likely since these mint species’ metabolites with antioxidant characteristics have a high hydrogen donating capacity [27]. It is congruent with our results where Mex showed the highest reducing capacity and two discriminant potent antioxidants, such as the chlorogenic acid and the antheraxanthin. These metabolites work as a hydrogen donator and electron donator, respectively. In addition, the botanical extracts of mint showed no significant difference in their ABTS radical inhibition capacity value, reporting activities between 48 and 60%, which is similar to that referenced by [29], who found a value of 40.2 ± 0.2%.

The higher Rosmarinic Acid (RA) content in the Eg sample could be due to genotype-environment interaction effects. *Mentha spicata* L. materials planted in the regions of Egypt have undergone some breeding process (natural or artificial) that favors the accumulation of rosmarinic acid, as reported by [30]. The RA can be further enhanced by the breeding process used in Central and Latin America (CLATAM), which is primarily clonal. Given the impossibility of the plant to naturally complete its flowering stage in these regions, it substantially reduces its diversity by sexual reproduction [31,32]. Additionally, as an emerging product in CLATAM, the production conditions are not fully optimized, which favors the bioaccumulation of target metabolites in this species. Consequently, abiotic factors, such as agronomic practices, cultivation conditions (open field or protected agriculture), and fertilization may influence the concentration of RA.

Some of the metabolites found in our spearmint extracts have also been previously reported for their antimicrobial activity on *S. aureus* and *S. aureus* methicillin-resistant but in cocoa extracts. Similarly, spearmint-related metabolites also had significant activity against gram-positive bacteria, such as *S. aureus*, in coffee extracts (Martínez-Tomé et al.) [33,34]. Among these metabolites, chlorogenic acids have been reported to have an antimicrobial effect on both gram-positive and gram-negative bacteria, exhibiting bactericidal activity against *P. fluorescens* and *S. aureus* but no sporicidal activity against *B. cereus* and *C. sporogenes* [34,35]. In this case, byproducts containing chlorogenic acids are considered promissory useful sources for pharmaceutical, cosmetic, and food industries [36]. In our results, the positive effect of spearmint extracts on Gram-positive bacteria may be due to their bacterial membrane composition, which is more easily permeable than Gram-negative bacteria membranes. Gram-positive membranes are more likely penetrable, generate complexes with extracellular proteins and soluble proteins quickly, and binding to the bacterial DNA [22].

The chemical and pharmacological analysis of the spearmint extracts (commercial materials from different sources) allows the determination of the potential of the samples for the possible development of ingredients or products of interest in the cosmetic, food, and pharmaceutical industries. In this way, spearmint’s chemical properties and pharmacological characteristics must be considered simultaneously. The accumulation of phenolic compounds in plant tissues is a distinctive feature of environmental stress. Polyphenolic compounds help plants cope with multiple biotic and abiotic stresses, such as drought, heavy metals, salinity, temperature, and ultraviolet light. As these conditions are variable in different origins, that could shape the metabolic profile of each commercial material. Furthermore, a specific metabolic profile varies significantly depending on the extract preparation, particularly when it comes to industrial processing for different industries [9].

The causality analysis shows a positive influence on all antioxidant activities, the total polyphenols, and chlorogenic acid, which is not surprising given the multiple reports of these compounds as inhibitors of oxidative radicals [26]. On the contrary, the compounds such as salvigenin and esculetin show a decrease in antioxidant biological activities. This is in agreement with the studies where salvigenin and esculetin present low antioxidant activity. Indeed, salvigening is reported with lipid-lowering and mitochondrial-stimulating activity, while esculentin shows anti-inflammatory, anti-apoptotic, anticancer, antidiabetic, and neuroprotective activities [37,38]. The antimicrobial causality analysis shows that the compounds with the most significant influence on the MRSA and *E. faecalis* inhibition are chlorogenic acid and esculentin, which are metabolites that are reported as compounds with an antibiotic capacity on gram-positive bacteria [38,39].

We found that esculetin contributed to the separation of the Col sample, and this sample also showed the highest SARM percentage inhibition. Therefore, we could propose that spearmint from Colombia could serve in products targeting antioxidant activities, UV protection, and antimicrobial activities. In contrast, the rosmarinic acid contributed to the separation of the Eg sample from the two other origins. It also agreed with the rosmarinic content for the Eg sample. Consequently, based on the health potential given for rosmarinic acid, the Eg spearmint could be used for pharmaceutical uses. Finally, a higher and varied number of metabolites allowed separate Mex samples from the other commercial samples, including the potent antioxidant chlorogenic acid and antheraxhanthin.

Furthermore, the Mex samples showed the highest inhibition percentage for *E. fecalis* and the highest polyphenol and antioxidant activity by FRAP. In this regard, the Mex spearmint samples have a higher potential to serve in both pharmaceutical and cosmetic uses [40]. Moreover, spearmint extracts have great potential in food industries since an increased awareness among consumers has increased the consumption of medicinal plants, mainly as natural antioxidants [41]. The applications of mint in food have been described as flavoring and preservative due to its antimicrobial properties [12]. Some applications have reported mint for fish and seafood, confectionery, chewing gum, and the cheese industry [3]. Altogether, and thanks to the metabolic composition and the biological characterization of commercial materials, the mint market can strengthen the decision making and apply spearmint for uses in cosmetics [42,43], pharmaceutics [44,45], and food [46].

## 4. Materials and Methods

### 4.1. Mentha spicata *L.* Commercial Materials

For the present investigation, three foliar commercial materials of dry and ground *Mentha spicata* L. were selected, which presented an active agro-industrial market for their commercialization in food companies or natural products in Colombia. For this, three different origins were selected: Eg (mint produced in the El-Fayun region, Egypt), Mex (mint produced in Atlixco, Mexico), and Col (mint produced in El Retiro-Antioquia, Colombia). The average climatic conditions of the production areas were obtained using the CHELSA database (https://chelsa-climate.org/) (accessed on 30 March 2022) and are consolidated in the Supplementary Material Appendix A.

### 4.2. Biological Activities and Chemical Asssays

#### 4.2.1. Preparation of Botanical Extracts of *Mentha spicata* L.

For the preparation of the botanical extracts, the methodology optimized by Sierra et al. [47] was used, for which 4 ± 0.05 g of each of the commercial mint materials (dry and ground plant material) were weighed, and a 30 mL of 80% HPLC-grade ethanol extraction solution in type I water was added. Subsequently, they were subjected to sonication with a frequency of 37 Hz, 30 ± 5 °C for 30 min (Elma P60H, Singer, La Vergne, TN, USA, EE. UU.) and then centrifugation at 8000 rpm for 20 min (Sorvall, Thermo Scientific, Waltham, MA, USA, EE. UU.) was applied to recover the supernatant fractions. All extraction assays were performed in triplicate.

#### 4.2.2. Determination of Total Phenolic Content (TPC)

The Folin-Ciocalteu method was carried out to determine the TPC content of each extract [48]. Gallic acid, in a dynamic range of 10 to 100 μg/mL, was used as a reference standard. Then, 25 μL of the extract was mixed with 125 μL of Folin-Ciocalteu’s reagent (1:10), both of which were diluted in distilled water. The mixture was shaken and incubated in darkness for 5 min at room temperature, followed by the addition of 100 μL of Na_2_CO_3_ (7.5% *w*/*v*). After 60 min of incubation at room temperature in the dark, absorbance readings were performed at 765 nm using a Synergy HT multimodal microplate reader (Biotek Instruments, Inc.; Wonooski, VT, USA). The total polyphenol content was calculated using a calibration curve with gallic acid. The results are expressed as milligrams of gallic acid (GA) per 100 g of extract, as appropriate (mg GA/g extract). All measurements were performed in triplicate.

#### 4.2.3. DPPH Radical Scavenging Capacity Assay

According to Brand-Williams et al. [49], the ability of Mentha leaf extracts to scavenge the DPPH (2,2-diphenyl-1-picrylhydrazyl) radical was determined. The reaction was composed of the Aliquots (10 µL) of the leaf extract and 990 µL of the DPPH standard solution. The absorbance was determined at 517 nm after 30 min in the dark (Biotek Instruments, Inc.; Wonooski, VT, USA). The results were expressed as millimoles of the Trolox Equivalent per 100 g of fresh Mentha (mmolTE 100 g^−1^).

#### 4.2.4. Reducing Power Assay (FRAP)

The reduction ability was measured using the methods of Benzie and Strain [50] with some modifications. Aliquots of 50 µL of the sample extract were mixed with 50 µL of an acetate buffer, pH 3.6, and adjusted to 1000 µL with a FRAP solution (FeCl_3_, TPTZ (Tripyridyl-s-triazine) in HCl 40 mM). The increase in absorbance was measured at 590 nm (Biotek Instruments, Inc.; Wonooski, VT, USA). The FRAP values were expressed as milligrams of the Ascorbic Acid Equivalent per 100 g of fresh Mentha (mgAAE 100 g^−1^).

#### 4.2.5. ABTS Radical Scavenging Capacity Assay

Radical scavenger activity against the stable radical, ABTS, was measured according to Mesa-Vanegas et al. [51]. An aliquots of 10 µL of the sample extract and 990 µL of the standard ABTS were mixed. The absorbance was determined at 732 nm after 30 min in the dark (Biotek Instruments, Inc.; Wonooski, VT, USA). The results were expressed as millimoles of the Trolox Equivalents per 100 g of fresh Mentha (mmolTE 100 g^−1^).

#### 4.2.6. Quantification of Rosmarinic Acid (RA)

Rosmarinic acid was identified and quantified using a methodology previously established in the laboratory [47]. It is based on reverse phase HPLC/DAD chromatographic analysis (Agilent Technologies, Palo Alto, CA, USA). Chromatographic separation was performed with a Zorbax SB RRTT C18 column (50 mm × 4.6 mm with 1.8 µm of particle size) of fast resolution and high performance using the mobile phase previously filtered through 0.45 µm nylon membrane filters and degassed using an ultrasonic bath before the analysis. The isocratic elution curve consisted of water with formic acid 0.5% (A) and acetonitrile (B), and the linear gradient used was as follows: 0 min, 16% B; 4 min, 16% B; 8 min, 20% B; 11 min, 40% B; 12 min, 45% B; 13 min, 50% B; 14 min, 60% B; and 15 min, 16% B, at a flow rate of 1.0 mL/min. The column temperature was maintained at 30 °C, and the injection volume of both the calibration curve of rosmarinic acid (CAS No. 20283-92-5, purity 99%, European Pharmacopoeia reference, Europe) and the sample solutions was 5 µL. The wavelength was set at 329 nm to monitor the chromatographic profile. All measurements were performed in triplicate [47].

### 4.3. Antimicrobial Activity in Commercial Materials of Mentha spicata *L.*

Hydroalcoholic extracts were prepared as described above to evaluate the antimicrobial activity of three commercial materials of *Mentha spicata* L. These extracts were used for the bacterial inhibition test following the protocol published by Balouiri et al. [52]. The inhibitory effect from the extracts was carried out against opportunistic pathogens, such as *Enterococcus faecalis* (ATCC 19433), *Escherichia coli* (ATCC 25922 and XL1 blue), *Klebsiella pneumoniae* (ATCC 700603), *Pseudomonas aeruginosa* (ATCC 27853), *Staphylococcus aureus* Methicillin-resistant (MRSA) (ATCC 43300), and *Streptococcus pyogenes* (ATCC 19615). The bioassay was realized in a 96-well microtiter; the wells were adjusted to 200 µL as the final volume. In addition, each treatment consisted of a 170 µL LB culture medium (50% Bactotryptone, 25% yeast extract, 35% NaCl, and diluted in distilled water), 20 µL of each extract, and 10 µL of the respective overnight bacterial culture. It was used as a negative control (hydroalcoholic solvent) and as a positive control (Gentamicin 40 µg*L^−1^). Afterward, the bacteria cultures were incubated at 37 °C, and the growth was monitored by measuring the absorbance at OD 600 until 24 h. Each treatment was carried out in triplicate, and the percentage of the inhibition was calculated by using the following equation (Control OD − (Sample OD/Control OD)) × 100.

### 4.4. Metabolic Profiling of Spearmint Based on HPLC-MS and HPLC-MS/MS

An ACQUITY H-CLASS UPLC-Xevo G2-XS-QTOF (Waters, Herts, UK) kit was used to obtain the undirected metabolic profile. The MS system was operated using electrospray ionization in positive mode, using an ACQUITY UPLC BEH C18 column (100 mm × 2.1 mm, 1.7 µm particle size) as the stationary phase. The working temperature was 55 °C, with a capillary voltage of 5.5 kV and a cone voltage of 30 V. The mobile phase used was a binary gradient of ultrapure water + 1% formic acid (solution A) and acetonitrile + 0.1% formic acid (solution B). The elution began with a linear gradient, starting with A:60%–B:40% for 17.5 min, followed by B:100% from 17.5 min to 25 min, and ending with the gradient A:60%–B:40% until the end at 30 Minutes. The flow rate was 0.250 mL/min, and the injection volume was 5 µL.

The data was processed with MS-DIAL V. 4.70 (Yokohama, Japan) [53]. For the peak detection, a weighted smoothing average algorithm was used as linear, setting the smoothing level to 1 scan, the minimum peak width at 3 scans, and the minimum peak height at 1000 amplitude. For the alignment for the peaks, we used 0.1 min in the RT tolerance and 0.025 Da in the MS1 tolerance. Subsequently, the peak areas obtained were normalized to the sum of the total ion chromatograms, logarithmically transformed, and scaled from Pareto, and the missing values were imputed using the baseline value and the signals that were not present in 70% of the samples were removed.

Statistical analysis was performed with MetaboAnalyst 4.0 software (Montreal, QC, Canadá) [54] and then by a Principal Component Analysis (PCA). Then, a PLS-DA was carried out to determine the representative characteristics that present the greatest variation among the mint samples. The top six VIPs with values greater than 2.0 were selected to be considered as potential biomarkers of the differences between the samples. For the data annotation, an automatic annotation was first performed using MSDIAL with the Fiehn/Vaniya Natural Products Library. MS and RT were used to match in the compound annotation. Fragmented VIP metabolites were compared on their precursor ion with data obtained from the Metlin, PlantCyc, and MoNA bases, using a threshold of 20 ppm. Additionally, manual curing was performed using a base of in-house data built from the mint reports.

#### Metabolites Annotation

For the metabolite annotation process, extraction of the experimental spectra was performed using MZmine software (Okinawa, Japan). After that, a first analysis was performed, which consisted of a manual search in the database established for Mentha. For those that could not be annotated, an automatic screening process was performed, in which the experimental spectra were compared with several compounds obtained from the Mass Bank of North America database, specifically a database for *Mentha spicata* built from this database. It should be noted that this comparison takes into account various adducts in order to identify more precisely the metabolite that best fits. To corroborate this information, a literature analysis was carried out to identify those metabolites that have been found in *Mentha spicata* L. in positive ionization and were obtained from liquid chromatography. Finally, with this list of compounds, the search for theoretical spectra was performed using the FOODB database and the simulation of the spectrum with CFM-ID using SMILES. With these spectra, an automatic comparison was performed, taking into account the mass and intensity to establish those metabolites that present the best fit with respect to the experimental one.

### 4.5. Causality Analysis between Metabolomic Profiles and Biological Activities

The determination of causal relationships between variables associated with functional activities and metabolites’ candidates of commercial materials of *Mentha spicata* L. used a multiple regression model adjusted with a linear polynomial of degree one (Equation (1)). In this model, each response variable (φ) was assumed as a determined functional potential. On the other hand, the relative intensities of the discriminant metabolites (VIP) were taken as fixed effect covariates in the proposed model [55]. The matrix was subjected to a Cochrane–Orcutt optimization process, where the classic least-squares procedure is modified to allow autocorrelation between successive residuals. In this case, both the autocorrelation standard (ρ) and the model parameters $\left(\beta_{k}\right)$ are determined interactively using between 30 and 50 iterations until the change in the derived value of each parameter, compared to the previous step, is less than 0.01, as proposed by Sun, Lang, and Boning (2021) for situations in which the model residuals are not independent.
(1)$$\varphi =\beta_{0}\left(1-\rho \right)+\beta_{1}x_{1}+\beta_{2}x_{2}+\ldots .\beta_{k}x_{k}$$
where:
φ=Related biological activity factor$\beta_{k}=\mathrm{Discriminating}\;\mathrm{adjustment}\;\mathrm{parameters}\;\mathrm{associated}\;\mathrm{with}\;\mathrm{the}\;\mathrm{cause}-\mathrm{effect}\;\mathrm{relationship}$$x_{k}=\mathrm{Discriminating}\;\mathrm{metabolites}\;\mathrm{associated}$ρ=Autocorrelation standard.

## 5. Conclusions

This work shows the first comparison of the attributes of the functional interest between *Mentha spicata* raw materials available in the international aromatic plant market, demonstrating the high heterogeneity in chemical variability and possible applications of the available raw materials. The results confirm that the characteristics of their chemical quality and antioxidant and antimicrobial activity show marked differences between the materials evaluated. The Col samples have potential as antioxidants and antimicrobials, probably for cosmetics, based on their percentage of inhibition SARM and esculetin. The Eg samples may work for pharma uses based on their rosmarinic content. Also, due to the metabolic diversity in the Mex samples, Mentha from this origin could have potential in varied industries, such as cosmetics and pharma, and is also supported by its antimicrobial activities. Finally, according to consumer awareness, spearmint extracts have great potential in food industries, mainly in which consumers may improve their health by increasing the intake of functional plant-based foods.

## Figures and Tables

**Figure 1 molecules-27-03559-f001:**
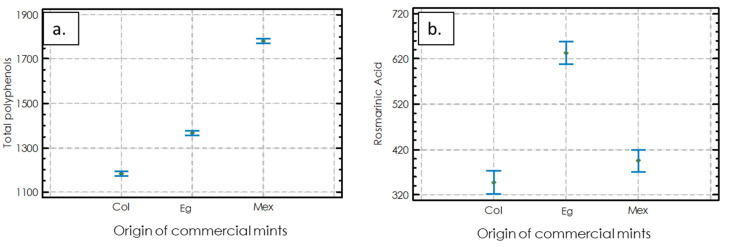
Chemical measurements in three commercial materials of *Mentha spicata L*. (**a**) Total polyphenol content (TPC) (mg of gallic acid (GA) per 100 g of extract); (**b**) rosmarinic acid concentration (mg rosmarinic acid (RA) per 100 g sample). Statistical analysis uses a one-way ANOVA (Tukey test, *p* < 0.05). Equal letters mean that there is no statistically significant difference.

**Figure 2 molecules-27-03559-f002:**
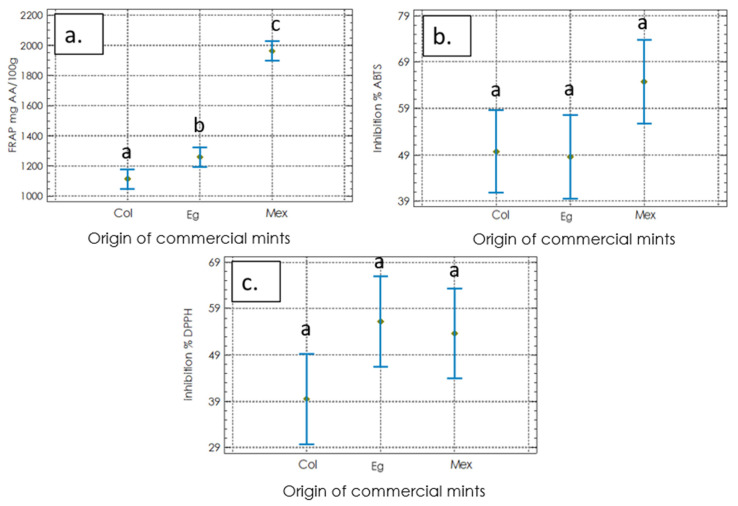
Antioxidant activities of three commercial materials from *Mentha spicata* L. (**a**) Ferric Reducing Antioxidant Power (FRAP activity) (mg Ascorbic Acid/ 100 g sample); (**b**) ABTS radical inhibition capacity (%); (**c**) DPPH radical inhibition capacity (%). Statistical analysis uses a one-way ANOVA (Tukey test, *p* < 0.05). Equal letters mean that there is no statistically significant difference.

**Figure 3 molecules-27-03559-f003:**
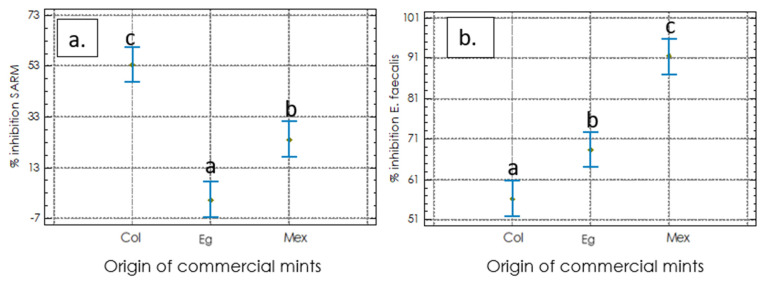
Antimicrobial activity of extracts from spearmint commercial materials against opportunistic bacteria; (**a**) percentage inhibition on Staphylococcus aureus Methicillin-resistant (SARM) (ATCC 43300); (**b**) percentage inhibition on Enterococcus faecalis (ATCC 19433). All data were collected in triplicate. One-way ANOVA using the Tukey test (*p* < 0.05) was used. The same letters mean that there is no statistically significant difference.

**Figure 4 molecules-27-03559-f004:**
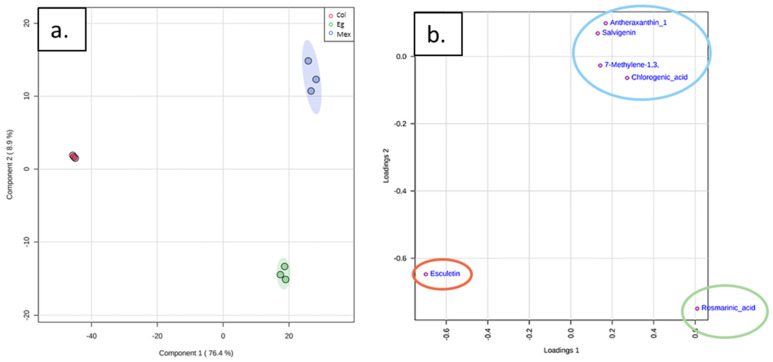
Multivariate analysis of discriminant metabolites in spearmint commercial materials. (**a**) PLS-DA for Mentha of three origins Col, Mex, and Eg. (**b**) Loading plot showing the six annotated metabolites for spearmint.

**Figure 5 molecules-27-03559-f005:**
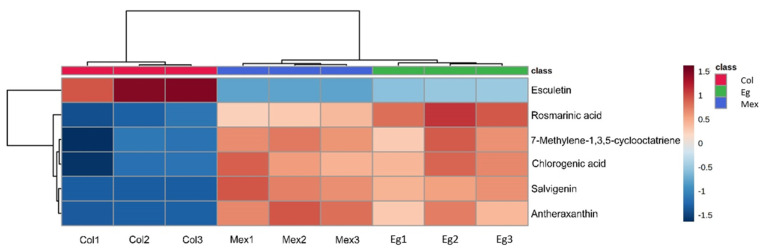
Heatmap showing expression patterns for six clustered metabolites from three commercial mints.

**Figure 6 molecules-27-03559-f006:**
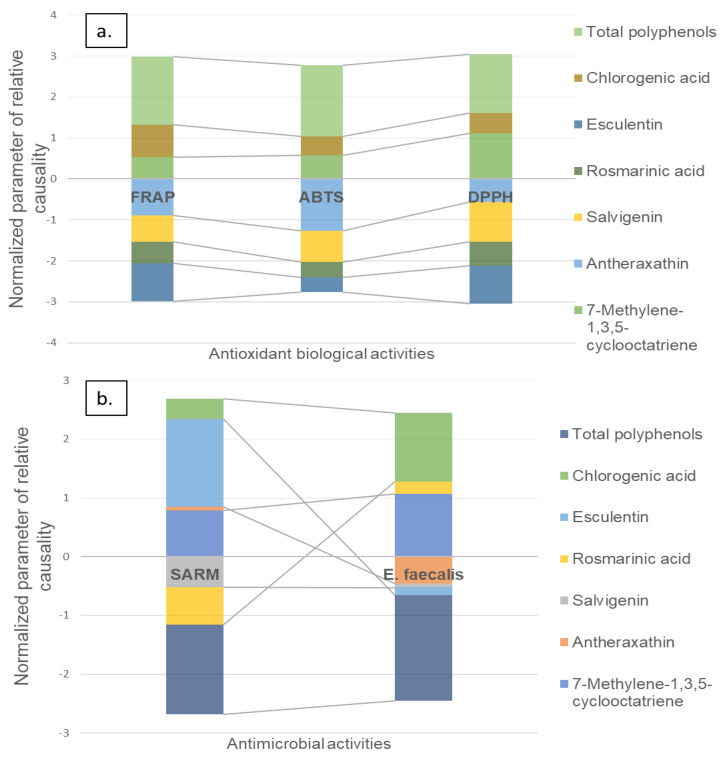
Causal relationships between antioxidant and antimicrobial activities and the main discriminating metabolites in *Mentha spicata* L. commercial materials of (**a**) antioxidant causality analysis; (**b**) antimicrobial causality analysis.

**Table 1 molecules-27-03559-t001:** List of annotated metabolites of spearmint (*Mentha spicata* L.).

No.	RT	*m/z*	Adduct	Molecular Formula	Anotación	Class
1	5.835	119.08387	[M + H]^+^	C_9_H_12_	7-Methylene-1,3,5-cyclooctatriene	Cycloparaffin
2	18.619	585.45056	[M + H]^+^	C_40_H_56_O_3_	Antheraxanthin	Carotenoid
3	18.609	329.1127	[M + H]^+^	C_18_H_16_O_6_	Salvigenin	Flavone
4	6.296	361.091	[M + H]^+^	C_18_H_16_O_8_	Rosmarinic acid	Caffeic acid ester
5	19.208	179.10556	[M + H]^+^	C_9_H_6_O_4_	Esculetin	Hydroxycoumarin
6	18.078	355.09079	[M + H]^+^	C_16_H_18_O_9_	Chlorogenic acid	Cinnamate ester

**Table 2 molecules-27-03559-t002:** Bioprospection of annotated metabolites in *Mentha spicata* L.

Compound	Food/Pharmacy Uses	References
7-Methylene-1,3,5-cyclooctatriene	Reports not found.	
Antheraxanthin	Skin care.	[16,17,18]
Salvigenin	Antiinflammatory and analgesic properties.	[19]
Rosmarinic acid	Antibacterial, antioxidant, anticancer, antiinflammatory, immunomodulatory, and coadjuvant activities in the treatment of cancer, diabetes, neuroprotective, and prevention of cognitive decline in Alzheimer’s disease.	[20,21,22,23]
Esculetin	Anti-inflamatory, anticoagulant, liver-protection, antidiabetic, antioxidant, antitumor, and UV-filters.	[24,25]
Chlorogenic acid	High potential antioxidant, hepatoprotective, cardioprotective, anti-inflammatory, antipyretic, neuroprotective, anti-obesity, antiviral, antimicrobial, anti-hypertension, and free radicals.	[26]

## Data Availability

Not applicable.

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
