# Peer review of "Towards Bioprospection of Commercial Materials of Mentha spicata L. Using a Combined Strategy of Metabolomics and Biological Activity Analyses"

_molecules, 2022, doi:10.3390/molecules27113559_

Round 1

Reviewer 1 Report

In line 34, spelling of ana-lyses should be corrected.

Line 60, 'acid' is missing in all the names of the phenolic acids. Syringic acid has been mentioned twice. Omit one.

Why the authors have chosen Mentha sp. from one country from North and South America respectively? Why Europe? No specific regions? Try to clarify  why contrasting origins?

Line 78 Spicata is the specific epithet?

Bio inspired product? Have the authors developed any product out of this study?

Data driven data - will be better if it is written as based on data analysis and interpretation.

Replace abbreviation like Col and EU.

From Line 81-82, The content of rosmarinic acid (RA) was higher in EU followed by Mex and next by Col samples. EU showed a significant difference in RA compared to the other two origins. What could be reason for this change in RA content. Explanation is needed,

In line 140 and 142, RY and Q2 should be written properly. 

What do the authors mean by Rosmarinic acid 1, 2, 3 and 4?

Likewise antheraxanthin 1 and 2 ? Are they not same compounds?

In Figure 5, EU, Mex And Col classes are not clustered? Data should be checked to get a better heatmap. 

in line 191, anteraxanthin, check spelling

Line 197 to 200 better explanation is needed. Could not get any clear idea.

Write it explicitely.

line 206, english correction  is needed.

Table 2. NO reported. English correction is needed.

Table 2 immunomodulatory and health enhancing activities for
cancer, diabetes and neuroprotective against Alzheimer . Verify with literature studies. Wrong english 

Table 2 High otential antioxidant.  free radicals. Could not understand. correction is needed. The authors have written the manuscript very hastily. not well presented.

Line 222 to 224, could not understand what the author wants to tell. Full of incorrect english. Check again.

Line 224, Presence of monoterpenes and monoterpenoids. Monoterpenes and Monoterpenoids are same or not?

LIne 225, cha-racteristic. Check and and correct.

The authors have used commercial samples from  agro-industrial market. Have you checked the adulteration of the commercial samples? How have you determined the quality control of the samples.

Line 296, delete an from an 80% HPLC-grade ethanol 

Line 311. All measurements were performed in triplicate. g of gallic acid (GA) per 100g of extract...... must write properly.

Line 322, 1000µ with FRAP solution...... unit should be written correctly.

Line 540 Bioactivides spelling mistake

Line 569, 575, Journal name is missing

Line 586, pg no. is missing

I don't find any novelty in this work. Kindly try to include.

Author Response

Response to reviewer

Manuscript ID: Molecules-1707227

Manuscript title: Towards bioprospection of commercial materials of Mentha spicata L. using a combined strategy of metabolomics and biological activity analyses

Dear Prof. Dr. Farid Chemat, Ph.D.

Editor-in-Chief

Molecules

On behalf of all the co-authors of this document, I would like to thank the work and time of the jurors in reviewing in detail and contributing to the improvement of the quality and clarity of our work. We find the reviews helpful and have responded to each of your comments in detail. Based on the reviewer's comments, we made modifications to the original manuscript and carefully revised it.

We detail our responses to each of the comments below, and we believe that the manuscript has been greatly improved by your input. We hope that this contribution will be to the liking and approval of the reviewers for publication in their journal.

For the answers and modifications, we refer in each of them to the line numbers. In addition, the language of the document was thoroughly reviewed and corrected as requested, a process carried out by an English language expert.

Response to reviewer 1 comments:

  • In line 34, spelling of ana-lyses should be corrected.

R: Reviewer's suggestion accepted and corrected in document text.

  • Line 60, 'acid' is missing in all the names of the phenolic acids. Syringic acid has been mentioned twice. Omit one.

R: Reviewer's suggestion accepted and corrected in document text.

  • Why the authors have chosen Mentha sp. from one country from North and South America respectively? Why Europe? No specific regions? Try to clarify  why contrasting origins?

R: The paragraph has been supplemented with the reviewer's suggestion as follows:

"These materials from different countries of origin were chosen because they are the ones with the highest proportion of use by food and cosmetic companies currently present in the Colombian market. Its proper use could generate high rates of productive performance, efficiency and homogeneity of the products obtained, impacting direct economic costs."

Additionally, the word contrasting origins is eliminated, it is made explicit that the reason for choosing mentha suppliers worldwide is due to their entry into the country where the research is carried out, where there is currently no significant presence of Mentha spicata materials from other South American countries or the United States.

  • Line 78 Spicata is the specific epithet?

R: Yes, its specific epithet is spicata and throughout the text it is named as Mentha spicata L.

  • Bio inspired product? Have the authors developed any product out of this study? Data driven data - will be better if it is written as based on data analysis and interpretation. Replace abbreviation like Col and EU.

R: The reviewer's suggestion is accepted, modifying the intentionality of the wording as follows:

“Our results provide decision-making tools based on analysis and data interpretation for the selection of specific raw materials of Mentha spicata L. for the future generation of products.”

In addition, the abbreviations have been replaced as recommended by both reviewers.

  • From Line 81-82, The content of rosmarinic acid (RA) was higher in EU followed by Mex and next by Col samples. EU showed a significant difference in RA compared to the other two origins. What could be reason for this change in RA content. Explanation is needed.

R: The content of the text is complemented with the following explanation:

“The results of the Rosmarinic Acid (AR) concentration can be explained by a possible genotype-environment interaction effect, since it is likely that the Mentha spicata L. materials planted in the regions of Egypt have undergone some breeding process (natural or artificial) that favors the accumulation of rosmarinic acid as those reported by [1], this is enhanced by the plant multiplication technology used in Latin America and Central America, which is mostly clonal, given the impossibility of the plant to complete its flowering stage naturally in these regions, which generates a substantially lower diversity than in areas with sexual reproduction [2,3]. Additionally, being an emerging product in Central America and Latin America, production conditions are not fully optimized for the bioaccumulation of metabolites of interest in this species, so abiotic factors such as agronomic practices, cultivation conditions (open field or protected agriculture), fertilization or unfavorable anthropic factors can influence the concentration of AR.” (Line 239:250)

  • In line 140 and 142, RY and Qshould be written properly. 

R: Reviewer's suggestion accepted and corrected in document text.

  • What do the authors mean by Rosmarinic acid 1, 2, 3 and 4? Likewise antheraxanthin 1 and 2 ? Are they not same compounds?

R: As we are interested in finding markers that could discriminate between origins, the repeated annotated metabolites were removed to better clarity in the text. Also, it made the article more consistent with the causality analysis.

  • In Figure 5, EU, Mex And Col classes are not clustered? Data should be checked to get a better heatmap. 
  1. We appreciate the reviewer comments. Indeed, each sample group is clustered. From the heatmap we can appreciate that samples from each origin groups together. It means, the three samples from EU group together, same for Col and Mex.
  • in line 191, anteraxanthin, check spelling.

R: Reviewer's suggestion accepted and corrected in document text.

  • Line 197 to 200 better explanation is needed. Could not get any clear idea. Write it explicitely.

R: This paragraph was deleted and replaced by:

The results of the causality analysis indicate that in general it is possible to obtain predictive capacity on the antioxidant and antimicrobial activities of samples of Mentha spicata extracts based on their discriminant metabolites. The above, using modified polynomial models with an adjusted R square greater than 98.5.

  • line 206, english correction is needed.

R: Reviewer's suggestion accepted and corrected in document text.

  • Table 2. NO reported. English correction is needed.

R: Reviewer's suggestion accepted and corrected in document text.

  • Table 2 immunomodulatory and health enhancing activities for
    cancer, diabetes and neuroprotective against Alzheimer. Verify with literature studies. Wrong english 

R: The text of the table has been modified as follows:

“Immunomodulatory and coadjuvant activities in the treatment of cancer, diabetes, neuroprotective and prevention of cognitive decline in Alzheimer's disease”

In addition, the references are added from number 17 to 20

  • Table 2 High potential antioxidant.  free radicals. Could not understand. correction is needed.

R: The text of the table has been corrected as follows:

“High potential antioxidant, hepatoprotective, cardioprotective, anti-inflammatory, antipyretic, neuroprotective, anti-obesity, antiviral, antimicrobial, anti-hypertension, free radicals.”

  • Line 222 to 224, could not understand what the author wants to tell. Full of incorrect english. Check again.

R: It is redrafted based on the reviewer's suggestions:

“With respect to antioxidant activities, our results of DPPH free radical inhibition capacity are consistent with reports from the Oman region (Arabian Peninsula), who reported 54.68% inhibition. Study in which the authors associated said inhibition, mainly with the presence of terpenes in extracts of spearmint”

  • Line 224, Presence of monoterpenes and monoterpenoids. Monoterpenes and Monoterpenoids are same or not?

R: It is replaced by the term "terpenes"

  • LIne 225, cha-racteristic. Check and and correct.

R: Reviewer's suggestion accepted and corrected in document text.

  • The authors have used commercial samples from  agro-industrial market. Have you checked the adulteration of the commercial samples? How have you determined the quality control of the samples.

R: Colombian legislation does not require adulteration tests for this type of product. However, quality and homogeneity tests related to the particle populations of the ground material (coarse and fine particles), microbiological analysis and reception moisture content were carried out. However, direct contact was maintained with the suppliers of the different countries and a technical file, location and product information was requested to guarantee that it was Mentha spicata. For the Colombian mint, a visit was made to the producing farms and the morphological characteristics of the species were validated.

  • Line 296, delete an from an 80% HPLC-grade ethanol 

R: Reviewer's suggestion accepted and corrected in document text.

  • Line 311. All measurements were performed in triplicate. g of gallic acid (GA) per 100g of extract...... must write properly.

R: Reviewer's suggestion accepted and corrected in document text: “The results are expressed as milligrams of gallic acid (GA) per 100 g of extract, as appropriate (mg GA/g extract). All measurements were performed in triplicate.

  • Line 322, 1000µ with FRAP solution...... unit should be written correctly.

R: Reviewer's suggestion accepted and corrected in document text (1000µL).

  • Line 540 Bioactivides spelling mistake

R: Reviewer's suggestion accepted and corrected in document text.

  • Line 569, 575, Journal name is missing

R: Journal name completed: "molecules"

  • Line 586, pg no. is missing

R: The number of pages of the citation has been completed: "915"

  • I don't find any novelty in this work. Kindly try to include.

R: Studies regarding to the metabolic profile of commercial Mentha spicata form different origins have not been previously reported. Based on that, we decided to analyze metabolic and functionally commercial extracts of Mentha spicata. We hypothesized that the combined analysis and information could provide guidance to the development of origin-specific products targeting different industries.

Sincerely, on behalf of all authors,

Juan Camilo Henao Rojas

Associate Master Researcher

Corporación Colombiana de Investigación Agropecuaria- Agrosavia

Reviewer 2 Report

Introduction

line 58-61 – this statement needs citations

line 64-65 – please add citations of literature

Authors should add paragraph about using the advanced statistics like PCA, PLS-DA, CA, etc. for different approaches such as: 1) distinguishing samples of different geographic origins by chemical composition, or 2) distinguishing samples after processing at different parameters or 3) to do comprehensive overview of obtained samples characteristics after addition of functional ingreadient. Here I recommend some examples: https://doi.org/10.3390/molecules26061802  , https://doi.org/10.1111/ijfs.14697 , https://doi.org/10.3896/IBRA.1.53.4.09

Results

Concise and clear presentation of results with good statistics. Only on the figures you will have to change the abbreviation EU to EG.

Discussion

The authors need to complete the section with rosmarinic acid content discussion as well as commentary and discussion with literature about Causality analysis.

Materials and Methods

line 289 - El-Fayun region, Egypt is neither in Europe nor part of the European Union. Egypt is located in Africa, so authors must change in whole article, including abstract and figures, the place of origin of analyzed material (instead UE should be EG because for abbreviations authors uses names of country).

line 286-293 - the authors must specify how many packages of a given production batch from a specific origin were used for analyses. Furthermore, the authors should test at least 2-3 production batches from a given place of origin to exclude the influence of climatic changes at a given place of cultivation on the composition and properties of the material. Please specify in the article if the authors did so?

In all methodologies please add model name and producer of devices used for preparing extracts and taking measurements.

line 298, 370 - please explain and describe in manuscript what is “type I water”.

line 299 - did you perform sonication with cooling or without? please add such information

line 300 - Please let us know if you made one or several extractions from a single weight of material. If you made one, please describe studies that have shown that one extraction from a single weight is sufficient.

Reducing Power Assay (FRAP), ABTS radical scavenging capacity assay – please precise: when you write “sample” you mean the extract? if so, please write instead “extract” or “sample extract”.

line 383-392- please check the usage of all abreviations, expand the VIP abbreviation also. What’s important separate information related to statistical analysis from information related to identifying compounds for better for better clarity of message.

line 396 – please name this database established for Mentha; all latin names need to be writed in italic

Conclusions

The authors' conclusions are too general. There is a need to expand this section with conclusions based on statistics of samples of different origin and microbiological analyzes, causality analysis.

Author Response

Response to reviewer

Manuscript ID: Molecules-1707227

Manuscript title: Towards bioprospection of commercial materials of Mentha spicata L. using a combined strategy of metabolomics and biological activity analyses

Dear Prof. Dr. Farid Chemat, Ph.D.

Editor-in-Chief

Molecules

On behalf of all the co-authors of this document, I would like to thank the work and time of the jurors in reviewing in detail and contributing to the improvement of the quality and clarity of our work. We find the reviews helpful and have responded to each of your comments in detail. Based on the reviewer's comments, we made modifications to the original manuscript and carefully revised it.

We detail our responses to each of the comments below, and we believe that the manuscript has been greatly improved by your input. We hope that this contribution will be to the liking and approval of the reviewers for publication in their journal.

For the answers and modifications, we refer in each of them to the line numbers. In addition, the language of the document was thoroughly reviewed and corrected as requested, a process carried out by an English language expert.

Response to reviewer 2 comments:

  • line 58-61 – this statement needs citations

R: The suggestion proposed by the reviewer is accepted and references [9] and [10] are added.

  • line 64-65 – please add citations of literature

R: The reviewer's suggestion is accepted and the citations are included [11,12]

  • Authors should add paragraph about using the advanced statistics like PCA, PLS-DA, CA, etc. for different approaches such as: 1) distinguishing samples of different geographic origins by chemical composition, or 2) distinguishing samples after processing at different parameters or 3) to do comprehensive overview of obtained samples characteristics after addition of functional ingreadient. Here I recommend some examples: https://doi.org/10.3390/molecules26061802  , https://doi.org/10.1111/ijfs.14697 , https://doi.org/10.3896/IBRA.1.53.4.09

R: The reviewer's suggestion is accepted and the PLS-DA goals were amended in the lines [XX]

Results

  • Concise and clear presentation of results with good statistics. Only on the figures you will have to change the abbreviation EU to EG.

R: The code "EU" has been replaced by "Eg" in all figures and texts.

Discussion

  • The authors need to complete the section with rosmarinic acid content discussion as well as commentary and discussion with literature about Causality analysis.

R: It is included in the discussion section on rosmarinic acid from lines 236 to 247, and discussion regarding the analysis of causality between lines 297 and 311.

Materials and Methods

  • line 289 - El-Fayun region, Egypt is neither in Europe nor part of the European Union. Egypt is located in Africa, so authors must change in whole article, including abstract and figures, the place of origin of analyzed material (instead UE should be EG because for abbreviations authors uses names of country).

R: The authors apologize for the confusion, the code for the samples had been placed based on the site of the packer of the Mentha material from Egypt. The suggestion of the reviewer is accepted and appreciated and it is changed throughout the document by the code "Eg".

  • line 286-293 - the authors must specify how many packages of a given production batch from a specific origin were used for analyses. Furthermore, the authors should test at least 2-3 production batches from a given place of origin to exclude the influence of climatic changes at a given place of cultivation on the composition and properties of the material. Please specify in the article if the authors did so?

R: The reviewer's suggestion is accepted and the information requested on lines 292 and 293 is included.

  • In all methodologies please add model name and producer of devices used for preparing extracts and taking measurements.

R: The evaluator's suggestion is accepted and the information on the missing laboratory equipment used is complemented.

  • line 298, 370 - please explain and describe in manuscript what is “type I water”.

R: The evaluator's suggestion is accepted and the expression “Type I water” was modified to Ultrapure water.

  • line 299 - did you perform sonication with cooling or without? please add such information

R: The reviewer's suggestion is accepted and line 328 is complemented with the extraction temperature (30 ± 5 °C).

  • line 300 - Please let us know if you made one or several extractions from a single weight of material. If you made one, please describe studies that have shown that one extraction from a single weight is sufficient.

R: In line 325 the reference of the optimized methodology that was used for the preparation of the botanical extracts is cited.

  • Reducing Power Assay (FRAP), ABTS radical scavenging capacity assay – please precise: when you write “sample” you mean the extract? if so, please write instead “extract” or “sample extract”.

R: The reviewer's suggestion was applied and the word "extract" was changed to "sample extract"

  • line 383-392- please check the usage of all abreviations, expand the VIP abbreviation also. What’s important separate information related to statistical analysis from information related to identifying compounds for better for better clarity of message.

R: The VIP abbreviation is expanded. The text was edited for better clarity.

  • line 396 – please name this database established for Mentha; all latin names need to be writed in italic

R: The database was build in house using references for the genera Mentha. The word was modified to use the common name mint.

Conclusions

  • The authors' conclusions are too general. There is a need to expand this section with conclusions based on statistics of samples of different origin and microbiological analyzes, causality analysis.

R: The conclusion was expanded according to the reviewer’s comment in the lines

Sincerely, on behalf of all authors,

Juan Camilo Henao Rojas

Associate Master Researcher

Corporación Colombiana de Investigación Agropecuaria- Agrosavia

Round 2

Reviewer 2 Report

The authors did a lot of work and significantly improved the manuscript. The authors have addressed all comments and observations. They filled in missing information and expanded the manuscript where needed. They also corrected the name of the origin of one of the three mint samples tested.